



# Spatio-Temporal Graph Neural Networks for Power Prediction in Offshore Wind Farms Using SCADA Data

Simon Daenens[1, 2, 3], Timothy Verstraeten[1, 2, 3], Pieter-Jan Daems[1, 2], Ann Nowé[3], and Jan Helsen[1, 2]

[1]OWI-Lab, Vrije Universiteit Brussel, Pleinlaan 2, 1050 Elsene, Belgium
[2]Acoustics & Vibration Research Group, Vrije Universiteit Brussel, Pleinlaan 2, 1050 Elsene, Belgium
[3]Artificial Intelligence Lab Brussels, Vrije Universiteit Brussel, Pleinlaan 2, 1050 Elsene, Belgium

**Correspondence:** Simon Daenens (simon.daenens@vub.be)

**Abstract.** This paper introduces a novel model for predicting wind turbine power output within a wind farm at a high temporal resolution of 30 seconds. The wind farm is represented as a graph, with Graph Neural Networks (GNNs) used to aggregate selected input features from neighboring turbines. A temporal component is added by feeding a timeseries of input features into the graph and utilizing a hybrid GNN-LSTM model architecture. This approach sequentially extracts temporal features with the LSTM component and spatial features with the GNN component. Our model is integrated into a Normal Behavior Model (NBM) framework for analyzing power loss events in wind farms. The results show that both the Spatial and Spatio-Temporal GNN models outperform traditional data-driven power curve methods, with the Spatio-Temporal GNN demonstrating superior performance due to its ability to capture both spatial and temporal dynamics. Additionally, we illustrate the model's effectiveness in detecting and analyzing instances of reduced performance and its ability to identify various types of abnormal events beyond what is recorded in standard status logs.

## 1 Introduction

Wind energy plays a vital role in addressing the escalating global energy demand while aligning with sustainability goals to combat climate change. As countries worldwide strive to reduce greenhouse gas emissions and transition towards renewable energy sources, wind energy emerges as a clean and abundant resource capable of meeting a significant portion of our energy needs. Offshore wind farms offer promising opportunities to harness stronger and more consistent wind speeds compared to onshore locations, thereby contributing to a more resilient and sustainable energy infrastructure. As technology advances and costs continue to decline, offshore wind projects are becoming more economically viable and attractive investments for governments, developers, and energy consumers alike. Moreover, offshore wind farms have the potential to generate large-scale energy outputs, making them crucial contributors to achieving renewable energy targets and reducing dependence on fossil fuels.

Accurate estimation of potential power allows wind farm operators to optimize the operation of individual turbines and the entire wind farm. By understanding the potential power each turbine could generate under different wind conditions, operators can make informed decisions about operation settings and maintenance schedules. As wind conditions and turbine performance can fluctuate significantly, developing precise power prediction models becomes essential to navigating these complexities.



Building on this need for accurate power forecasts, it is equally important to utilize these predictions to identify power loss events - instances of reduced performance where the generated power falls below the potential output - within the wind farm. By comparing predicted energy output with actual performance, we can detect potential inefficiencies and operational issues.

In this context, this research addresses two key questions:

1. Can we leverage temporal and spatial patterns within a wind farm to improve power prediction accuracy?

2. Can we detect power loss events by comparing precise power predictions with actual turbine performance?

Potential power prediction methods can be integrated into a broader performance monitoring framework through the implementation of normal behavior modeling (NBM). The essence of NBM lies in training models to recognize normal behavior, enabling the identification of abnormal observations based on low model support. For instance, in the context of a regression-based NBM model, training on a dataset representing purely normal behavior enables the detection of abnormalities by identifying deviations from an expected residual of zero when presented with new test set observations.

Additionally, quantifying losses during power loss events enables stakeholders to assess the economic impact of turbine downtime or grid curtailments. High levels of curtailment can lead to increased wear and tear on turbines, reduced operational lifespan, and potential reliability issues (Robbelein et al., 2023). With curtailment strategies evolving to be more dynamic, the need to refine these strategies increases. By understanding the magnitude and frequency of the power losses during curtailments, operators can implement measures to minimize curtailment frequency or duration and make informed decisions about investments in grid upgrades or energy storage solutions.

Finally, the development of state-of-the-art wind farm control methods typically relies on models that capture the wake effect between wind turbines at a high temporal resolution (Verstraeten et al., 2021). Wake effects are highly non-linear and difficult to capture, and traditionally modeled using physics-based wake models that are often calibrated using real-world data (Van Binsbergen et al., 2024a, b, c). In this work, we propose an alternative to physics-based models for capturing wake effects and predicting the potential power of wind turbines operating in wakes, specifically designed for wind farm control applications.

The main contributions of this paper are the following:

– We developed a model to predict the potential power of all turbines within a wind farm at high temporal resolution (30 second) by modeling the wind farm as a graph and using Graph Neural Networks to aggregate selected input features from neighboring turbines and predict the power output across the wind farm.

– A temporal component was added by feeding a timeseries of the input features to the graph and using a hybrid GNN-LSTM model architecture to sequentially extract the temporal features using the LSTM and the spatial features with the GNN component.

– We integrate our power prediction model into a NBM framework and showcase the practical implications of turning power prediction methods into a methodology for analyzing power loss events.





## 2 Related Work

An example of a normal behavior model, consisting of an artificial neural network variant trained on an abnormality-filtered dataset is shown by Lyons and Gocmen (2021). The authors show its effectiveness in accomplishing the power performance analysis objective, and instances of over- and underperformance captured by the developed NBM network are presented and discussed. However, the NBM seemed to struggle to model power production at higher wind speeds and expressed a tendency to underestimate the wake effect at play within the farm. In another study by Bilendo et al. (2022), a different normal behavior model is introduced, leveraging a heterogeneous stacked regressor (HET-SR) algorithm. This algorithm learns from optimal power curve data to serve as a predictive model within their NBM framework. While their findings provide evidence on the effectiveness of the proposed method for accurate fault detection, employing more advanced prediction models that account for additional variables beyond wind speed could offer opportunities for a more comprehensive performance analysis beyond general fault detection.

Traditionally, wind power production is governed by the power curve, a simple relationship that gives the expected power at different wind speeds. However, it cannot provide a precise forecast of wind power as it fails to accurately model the non-linear relationship between wind speed and power output, especially in large wind farms where wake losses become more prevalent. Modeling these wake losses and the flow patterns within the farm has been an active area of research and various approaches have been devised to predict expected power in wake-affected wind farms. These methods can be categorized as either physics-based or data-driven (or a combination of the two).

Physics-based models aim to model the wind and wake flows within the wind farm based on prior knowledge about the physical behavior of the system. These models range from low to high fidelity, depending on the amount of detail they capture. Low-fidelity models are relatively fast, but neglect lots of details in modeling the wind flow. An example is the FLOw Redirection and Induction in Steady State (FLORIS) model developed at NREL, a control-focused wind farm simulation software incorporating steady-state engineering wake models into a Python framework (NREL, 2024). On the other hand, high-fidelity models describe the flow in detail based on the 3D Navier-Stokes equations and use large-eddy simulations (LES) to accurately resolve the turbulent flow structures, but they are limited by their high computational cost. Examples of high-fidelity models are SOWFA (NREL, 2012) and PALM (Raasch and Schröter, 2001).

Data-driven models construct relationships between the inputs and outputs based on statistical or machine learning models, without prior knowledge about the physical behavior of the process. Recent advancements in deep learning and big data resulted in a growing interest towards wind farm power prediction using deep learning methods. For instance, Lin and Liu (2020) provide a comprehensive overview of prior studies employing deep learning methods and introduces a predictive model using deep learning in conjunction with high-frequency SCADA data. Similarly, Lyons and Gocmen (2021) delve into power prediction tasks using deep learning, utilizing high-frequency SCADA data and integrating local information from neighboring turbines to enhance predictive accuracy. Moreover, the significance of spatiotemporal factors influencing wind power generation is addressed in Zhang et al. (2021) and Yin et al. (2021), employing specialized model architectures such as Convolutional Neural Networks (CNNs) and Long Short- Term Memory networks (LSTMs) to extract spatial and temporal feature informa-





tion. Finally, Daenens et al. (2024) developed a turbine-level power prediction model, incorporating high-frequency SCADA data from neighboring turbines into a prediction model for potential power based on a hybrid CNN-LSTM model architecture, by organizing the input data in a grid structure based on the wind farm layout.

A different approach to represent the spatial correlations between wind turbines in a wind farm is the use of graphs. Graphs can be used to represent complex data that is not necessarily ordered in a grid-like structure, but instead can be modeled by arbitrary graphs consisting of nodes and edges. With this approach, Graph Neural Networks (GNNs) can be employed for the prediction of wind farm power output. For example, Bleeg (2020) introduced GNNs as surrogate models for steady-state Reynolds-Averaged Navier-Stokes (RANS) simulations, providing a more computationally efficient alternative to traditional RANS models while maintaining reasonable accuracy. More complex GNNs are explored in the work of Park and Park (2019) and Bentsen et al. (2022), where the former proposed a physics-induced GNN (PGNN) and the latter an attention-based GNN for the power prediction of individual wind turbines in a wind farm. Both these methods use synthetic wind farm data simulated using the FLORIS model. Yu et al. (2020) developed the Superposition Graph Neural Network, a spatio-temporal model that processes a timeseries of graphs to predict the power output of the wind turbines in a wind farm. This model was trained on data for four offshore wind farms sampled at 10-minute intervals.

## 3 Methodology

The data-driven methodology proposed in this work leverages high-resolution SCADA data to predict the potential power for each wind turbine within a wind farm. In this section, we will describe the key aspects of our modeling framework and we will show the design choices that were made to develop a unified framework that can be applied to any wind farm. The data as well as the preprocessing pipeline are described in detail in Sect. 3.1, and the different model architectures and characteristics are elaborated on in Sect. 3.3.

### 3.1 Data collection and preprocessing

We considered a wind farm in the Dutch-Belgian offshore zone for this study, consisting of over 40 turbines with >8MW rated power. Specifically, the signals obtained by the supervisory control and data acquisition (SCADA) system were used, spanning a two-year period from 2021 to 2022. The SCADA system records real-time information at a resolution of 1 Hz collected from sensors and control systems for monitoring and optimizing turbine performance and overall wind farm operations and is widely used in the offshore wind industry. As most contemporary wind farms have a SCADA system, a SCADA-based methodology is easily adaptable to other wind farms.

From the SCADA system, the data signals shown in table 1 were retained and used in our analysis. The input features to the prediction model are `wind_speed`, `wind_direction_sin`, `wind_direction_cos`, and `turbulence_intensity`. Wind speed is directly taken from the SCADA data and represents the wind speed at turbine hub height as measured by the anemometer located on the nacelle. Wind direction in our model is represented using sine and cosine transformations to account for its circular nature. This approach ensures a smooth representation of wind direction throughout its entire range and





prevents issues where a model might incorrectly interpret a large difference between wind directions close to 0 and 360 degrees. Turbulence intensity (TI) refers to the measure of fluctuations in wind speed over time, indicating the variability or instability
of the airflow. The turbulence intensity is calculated using Eq. (1).

$$TI = \frac{\sigma_{ws}}{\mu_{ws}} \tag{1}$$

where $\sigma_{ws}$ is the standard deviation of wind speed (ws), and $\mu_{ws}$ is the mean wind speed, both computed over a 10-minute interval centered around the 1-second data point.

The target of our prediction model is the `active_power` signal extracted from the SCADA data. Additionally, we include
two more SCADA signals: `rotor_speed` and `pitch_angle`, which are utilized for filtering normal operational behavior across our training, validation, and test datasets.

**Table 1.** Overview of Data Signals and Input Features

| Signal Name | Description | Data source | Usage |
|---|---|---|---|
| wind_speed | The wind speed as measured by the anemometer located on the turbine nacelle | SCADA | Model Input |
| wind_direction | The wind direction as measured by the wind vane located on the turbine nacelle | SCADA | Calculation of input features |
| wind_direction_sin | The sine of the wind direction | Calculated | Model Input |
| wind_direction_cos | The cosine of the wind direction | Calculated | Model Input |
| turbulence_intensity | The turbulence intensity as calculated by formula 1 | Calculated | Model Input |
| active_power | The active power produced by the wind turbine | SCADA | Model Target |
| rotor_speed | The rotational speed of the wind turbine blades | SCADA | Data Filtering |
| pitch_angle | The angle of attack of the wind turbine blades relative to the incoming wind | SCADA | Data Filtering |

The steps to prepare the raw data for the training, validation, and testing of our prediction model are outlined in Table 2.

First, the raw SCADA data with a sampling frequency of 1 Hz was resampled to 30-second averages as a trade-off between high temporal resolution, acceptable noise levels, and computational costs. Next, the data was annotated with control conditions
to be used when filtering for normal behavior. As mentioned previously, to create an NBM that can accurately detect abnormal behavior and predict potential power output under any conditions, the model must be trained on data representing normal system behavior. The filtration method employed in this study is physics-based and uses properties of the power curve based on the IEC standard to annotate steady-state control conditions. Using this approach, different control regimes can be identified, and outliers and abnormal behavior can be removed from the dataset. Additionally, to leverage the temporal patterns in the wind
flow through the wind farm, lagged values of the input features are incorporated into the model. Given the time-series nature of the data, it is important to maintain temporal continuity when lagged values are used as inputs. Consequently, a final filtering step was implemented to ensure that the lagged values utilized by the model are consecutive and there are no discontinuities due to missing data. Finally, the resulting dataset is partitioned into distinct subsets for model training, validation, and testing.





**Table 2.** Preprocessing Steps and Remaining Data Points

| Preprocessing Step | Initial Data Points | Remaining Data Points | Final Data Split |
|---|---|---|---|
| Raw SCADA Data | 39,118,140 | - | - |
| Resampling to 30s Averages | - | 1,303,938 | - |
| Retaining Normal Behavior | - | 262,614 | - |
| Consecutive Timestamps | - | 164,124 | - |
| Train, Validation, Test Split | | | |
| **Training Set** | - | - | 116,410 |
| **Validation Set** | - | - | 23,857 |
| **Test Set** | - | - | 23,857 |

## 3.2 Graph representation of the wind farm

With the datasets for training, validation, and testing of the prediction model defined, they have been transformed into a format suited for our model. Specifically, the wind farm is represented as a graph, and the selected input features from the SCADA data are converted into graph-structured data.

A graph $G$ is usually defined as a tuple of two sets $G = (V, E)$, where $V = \{v_1, v_2, \ldots, v_N\}$ and $E \subseteq V \times V$ are the sets of nodes and edges. To model a wind farm as a graph using this structure, each turbine in the wind farm is represented as a 150 node, and edges connect neighboring turbines. An edge $e_{ij} = (v_i, v_j) \in E$ between nodes $v_i$ and $v_j$ exists if these nodes have line-of-sight visibility and there are no obstacles (other turbines) blocking the direct view. This visibility check is performed using Algorithm 1.

---

**Algorithm 1** Visibility Check

---

1: **Input:** node1, node2, obstacles, tolerance

2: **Output:** True if node1 and node2 are in line-of-sight, False otherwise

3: Initialize a flag, *visible*, to True.

4: **for** each obstacle **in** obstacles **do**

5:     **if** obstacle $\neq$ node1 **and** obstacle $\neq$ node2 **then**

6:         Calculate the distance from node1 to the obstacle, $d1$.

7:         Calculate the distance from node2 to the obstacle, $d2$.

8:         Calculate the direct distance between node1 and node2, $d\_direct$.

9:         **if** $|d1 + d2 - d\_direct| <$ tolerance **then**

10:             Set *visible* to False and **break** out of the loop.

11:         **end if**

12:     **end if**

13: **end for**

14: **return** *visible*, indicating whether node1 and node2 have line-of-sight visibility.

---





Each node $v \in V$ can be associated with a vector of features $x_v \in X$, comprising the input features as defined previously. Similarly, a vector of edge attributes can be defined for each edge $e \in E$, containing a set of geometric features, such as the length and direction of each edge.

### 3.3 GNN Models

Graph Neural Networks (GNNs) (Scarselli et al., 2009) have emerged as powerful tools for learning on graph-structured data. They are particularly well-suited for tasks that involve node-level predictions, such as power prediction in wind farms. GNNs leverage the structure of the graph to learn rich node representations by iteratively aggregating information from neighboring nodes.

At the heart of GNNs is the concept of message passing, a process through which nodes in a graph communicate with their neighbors to update their representations (Gilmer et al., 2017). This iterative process can be described in two main steps: message aggregation and node update.

1. **Message Aggregation**: Each node aggregates messages from its neighbors. The nature of this aggregation can vary, but common methods include summation, mean, and max pooling of neighbor features.

2. **Node Update**: After aggregation, each node updates its own feature based on the aggregated message and its previous state. This update is typically performed using a neural network, such as a multilayer perceptron (MLP).

Mathematically, the $k$-th layer of a GNN can be expressed as:

$$\mathbf{h}_i^{(k)} = \text{UPDATE}\left(\mathbf{h}_i^{(k-1)}, \text{AGGREGATE}\left(\{\mathbf{h}_j^{(k-1)} : j \in \mathcal{N}(i)\}\right)\right)$$

where UPDATE and AGGREGATE are arbitrary differentiable functions (i.e., neural networks), $\mathbf{h}_i^{(k)}$ is the representation of node $i$ at the $k$-th layer, and $\mathcal{N}(i)$ denotes the set of neighbors of node $i$.

For the task of power prediction, we developed two models: a Spatial GNN and a Spatio-Temporal GNN. The Spatial GNN considers the node features only at the current timestep, whereas the Spatio-Temporal GNN also incorporates lagged values of the input features. Both models utilize the GENeralized Graph Convolution (GENConv) model proposed by Li et al. (2020). This model employs generalized message aggregators, pre-activation residual connections, and message normalization layers to ensure robustness in deep networks. Additionally, it can integrate edge features, which we believe contain crucial information about intra-farm wake behavior.

The Spatial GNN model consists of a node and edge feature encoder, followed by a series of graph convolutional layers (GCN layers), and concludes with a dense layer with sigmoid activation to normalize the predictions between 0 and 1. Both the node and edge encoders are two-layer multi-layer perceptrons (MLPs) with ReLU activations. The GCN layers are implemented using PyTorch Geometric (Fey and Lenssen, 2019), and comprise DeepGCNLayer objects that incorporate pre-activation residual connections. Each DeepGCNLayer employs layer normalization, ReLU activation, and the GENeralized Graph Convolution (GENConv) as the convolution operator.





The Spatio-Temporal GNN model first processes the time series of node features with an LSTM network. Similar to the
185 Spatial GNN, this block serves as the node feature encoder; however, in this case, the output of the LSTM network is passed
through the GNN model.

An overview of the methodology for power prediction is shown in Figure 1.

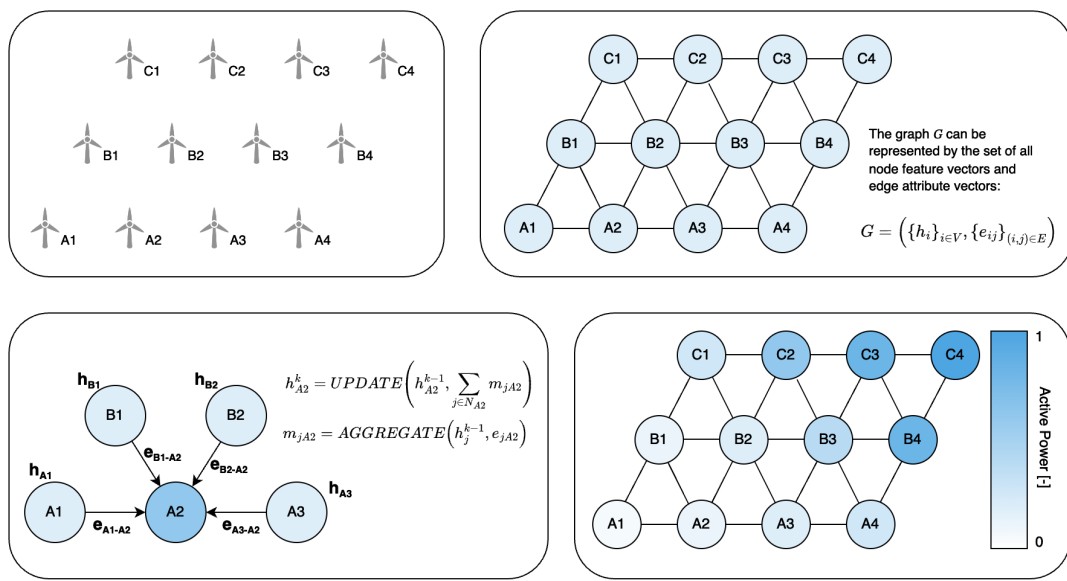

**Figure 1.** Overview of the proposed methodology for power prediction. The wind farm (a) is represented as a graph (b). The GNN learns
rich representations for each node in the graph using message passing (c). Finally, a power prediction is made for each node within the graph
(d).

### 3.4 Hyperparameter tuning

Hyperparameter tuning is crucial for optimizing a machine learning model's performance and ensuring it generalizes well to
190 unseen data. In this study, we used the hyperparameter optimization framework Optuna (Akiba et al., 2019). Optuna allows
users to dynamically construct the parameter search space, benefits from efficient sampling and pruning algorithms, and is easy
to set up.

Table 3 lists the hyperparameters that were tuned, along with their suggested ranges. After defining these ranges, Optuna
automatically searched for the optimal hyperparameters based on the objective function. Within the objective function, models
were trained with the selected hyperparameters and scored based on mean squared error (MSE).

Hyperparameter combinations that resulted in the lowest MSE for the validation dataset were retained and used to build the
final models.





**Table 3.** Hyperparameters and Their Considered Ranges

| Hyperparameter | Suggested Range |
|---|---|
| Encoding Channels | {8, 16, 32, 64} |
| Hidden Channels | {8, 16, 32, 64} |
| Number of Layers | {1, 2, 4, 8} |
| Number of LSTM Layers | {1, 2, 4, 8} |
| GNN Dropout | [0, 0.1] |
| Encoding Dropout | [0, 0.1] |
| Number of Epochs | [20, 100] |
| Batch Size | {64, 128, 256, 512} |
| Learning Rate (lr) | [1e-5, 0.5] (log scale) |
| Learning Rate Decay | [0.9, 1] |

## 4 Results & Discussion

In this section, the performance of the different models concerning the specified objectives is discussed. As mentioned in Sect.
3.1, SCADA data of an offshore wind farm comprising over 40 turbines has been used to train and evaluate the Spatial and
Spatio-Temporal GNNs described in Sect. 3.3.

### 4.1 Model performance during normal operation

In Table 4, the Mean Absolute Error (MAE) and the Mean Absolute Percentage Error (MAPE) are reported for each model. The
models are compared with a data-driven power curve method as a baseline. With the power curve method, the turbine's active
power is estimated from the power curve, created by analyzing collected data on wind speed and the turbine's actual power
production. This curve illustrates how the turbine responds to different wind speeds, enabling predictions of power generation
for varying wind conditions within the observed range.

**Table 4.** Performance Metrics. The MAE values have been normalized for confidentiality reasons. The lowest errors are highlighted to
showcase the model with the best predictive performance on each dataset.

|  |  | Power Curve | Spatial GNN | Spatio-Temporal GNN |
|---|---|---|---|---|
| MAE | Train | 0.0476 | 0.0287 | **0.0258** |
|  | Validation | 0.0481 | 0.0348 | **0.0311** |
|  | Test | 0.0478 | 0.0370 | **0.0333** |
| MAPE | Train | 12.717 % | 7.859% | **6.839 %** |
|  | Validation | 13.265 % | 9.916% | **8.516 %** |
|  | Test | 12.442 % | 9.872% | **8.647 %** |





The results indicate that both the Spatial GNN and the Spatio-Temporal GNN models significantly outperform the data-driven power curve method across all metrics and datasets. The superior performance of the GNN models emphasizes the

advantage of representing the wind farm as a graph. In this approach, each turbine not only considers its local measurements but also integrates information from its neighbors through message passing, enabling the model to capture complex spatial dependencies within the wind farm.

Specifically, the Spatio-Temporal GNN model exhibits the best performance, as evidenced by the lowest Mean Absolute Error (MAE) and Mean Absolute Percentage Error (MAPE) values in the training, validation, and test datasets. The Spatio-

Temporal GNN's ability to incorporate lagged values of the input features provides it with an additional temporal dimension, allowing it to detect and leverage trends over time. This capability is particularly beneficial for capturing dynamic changes in wind conditions, such as fluctuations in wind speed and direction, or for smoothing out the inherent noise in high-frequency SCADA data.

Comparatively, while the Spatial GNN also outperforms the power curve method, it falls short of the Spatio-Temporal GNN.

The Spatial GNN only utilizes input features at the time of prediction, missing out on the temporal trends that the Spatio-Temporal GNN can exploit. Therefore, while spatial awareness and the ability to model interactions between turbines are crucial, the inclusion of temporal dynamics further enhances predictive accuracy.

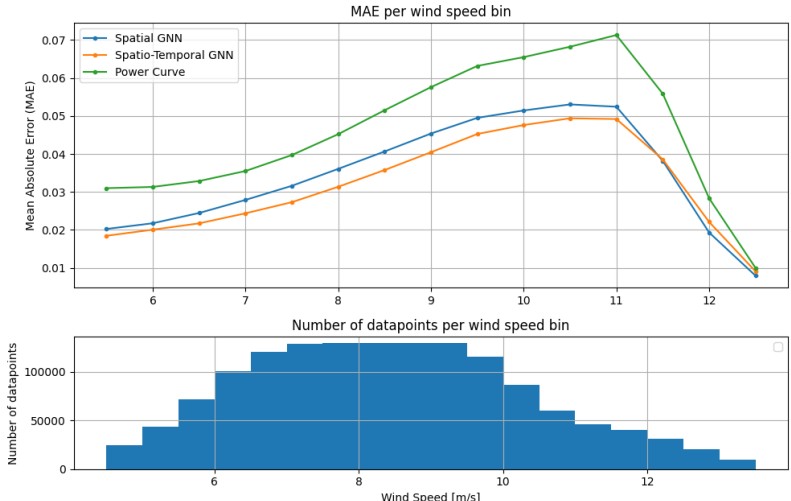

**Figure 2.** Mean Absolute Error (MAE) per wind speed bin for the two models and the power curve method. The plot at the bottom shows the number of data points per wind speed bin.

To provide a more granular analysis of our models' predictive performance, Figure 2 presents the Mean Absolute Errors (MAE) for different wind speed bins on the test dataset. While the previous analysis highlighted the overall performance of

each model across the entire dataset, this plot shows their performance across varying wind speeds.



The analysis shows a slight increase in MAE at higher wind speeds across all models, with a subsequent decrease in errors near the turbine's rated wind speed. This trend aligns with expectations, as higher wind speeds typically result in higher power outputs and consequently larger absolute errors. The observed pattern follows the inherent uncertainties of the power curve, but the models still achieve significantly higher accuracy.

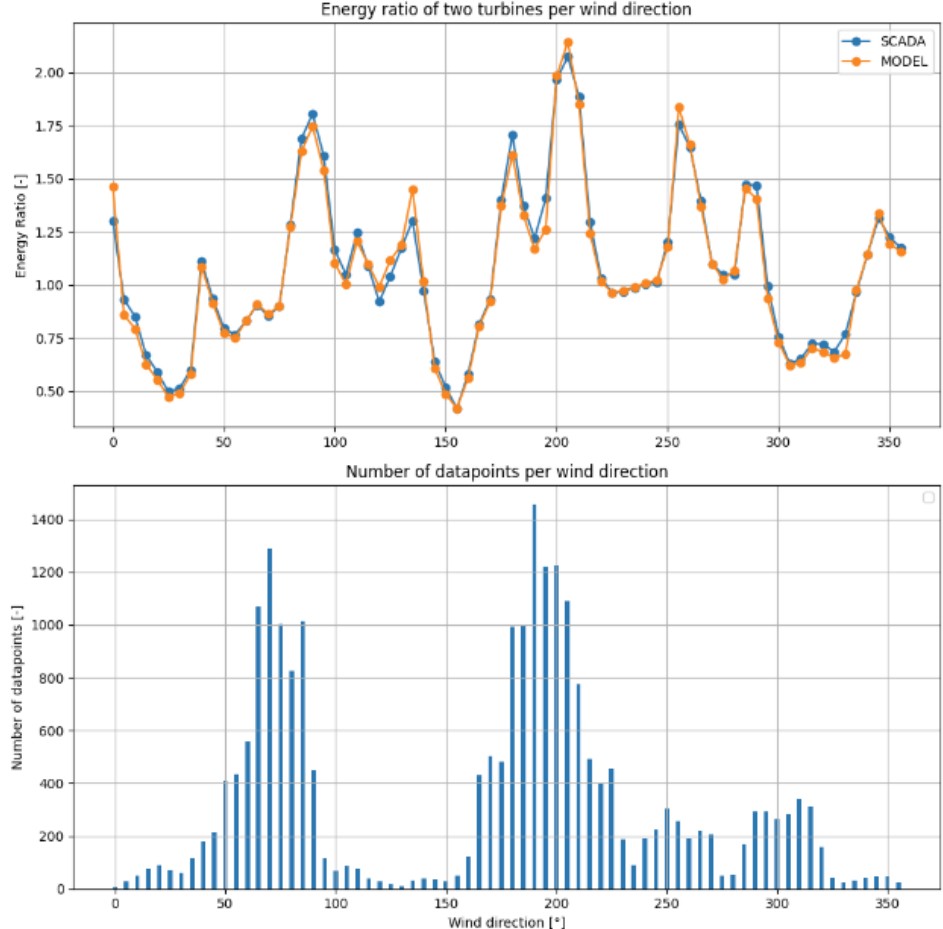

**Figure 3.** Energy ratio between two turbines per wind direction as predicted by the Spatio-Temporal GNN and from the SCADA data. The plot at the bottom shows the number of data points per wind direction bin.

An essential step in model validation involves comparing the model's predictions with historical data, specifically focusing on whether the individual turbine wake losses are accurately predicted. This can be achieved by analyzing the energy ratio between a test turbine and a reference turbine for each wind direction bin (Fleming et al., 2019; Doekemeijer et al., 2022). By examining the predicted energy ratio for each wind direction, we can gain deeper insights into how the model evaluates the impact of wake interactions within the wind farm. In an unbiased model, the energy ratio curves should align closely with those observed in the SCADA data.





The energy ratio $R_{Energy}$ is defined in Eq. (2) and represents the ratio of the sum of all the power measurements between a test turbine and a reference turbine, computed for each wind direction bin. In this equation, $P_{i,\theta}^{\text{Turbine 1}}$ and $P_{i,\theta}^{\text{Turbine 2}}$ are the observed powers of point $i$ in a given wind direction bin for the two test turbines, and $N$ is the number of points in this wind direction bin.

$$R_{\text{Energy}}(\theta) = \frac{\sum_{i=1}^{N} P_{i,\theta}^{\text{Turbine 1}}}{\sum_{i=1}^{N} P_{i,\theta}^{\text{Turbine 2}}}$$
(2)

Figure 3 illustrates the energy ratio curves for two test turbines within the wind farm. One turbine is positioned in the free flow relative to the dominant wind direction, while the other is located in a subsequent row behind the first turbine, thus experiencing the wake effect generated by the upstream turbine. These curves compare the energy ratio, per wind direction bin, as predicted by the hybrid model with the energy ratios derived from the SCADA data.

Our Spatio-Temporal GNN model demonstrates remarkable agreement with the energy ratios from the SCADA data across the full range of wind directions. Discrepancies are minor and primarily occur in wind directions that are underrepresented in the dataset. This suggests that the model's predictive accuracy is robust, even though it may be slightly less accurate in areas with sparse data.

Furthermore, the energy ratios align with intuitive expectations. For instance, in wind directions between 220° and 245°, 250 both turbines experience undisturbed wind inflow, leading to similar power outputs. As the wind direction shifts towards 250° to 260°, one of the test turbines becomes obstructed by another turbine, resulting in reduced power production. Consequently, the energy ratio increases, reflecting the higher power output of the turbine with unobstructed inflow compared to the blocked turbine.

This consistency between the model predictions and the intuitive understanding of turbine interactions under different wind 255 conditions shows the Spatio-Temporal GNN model's effectiveness in capturing the complex dynamics of wake interactions within the wind farm. The minor discrepancies in less common wind directions highlight areas for potential improvement but do not detract significantly from the overall performance of the model.

### 4.2 Predictability of abnormal events

Having quantified our models' accuracy on the test set containing normal behavior, the next step is to assess how the models 260 handle abnormal behavior. As discussed in the introduction, our objective is to integrate our power prediction model into a normal behavior modeling (NBM) framework. With our power prediction model trained to predict normal behavior, we propose a methodology to detect abnormal observations based on low model support.

In our methodology, low model support is defined as a deviation of the predicted power from the actual power that exceeds three times the standard deviation of the produced power for a given wind speed. The standard deviations are derived from a 265 data-driven power curve.

At low wind speeds, the standard deviation of the produced power is relatively low because the potential power output is low and more predictable. As the wind speed increases, the uncertainty of the potential power output also increases, leading




to a higher standard deviation. Approaching the rated wind speed, the potential power output converges to the rated power, reducing uncertainty and hence decreasing the standard deviation again.

This approach means that our NBM methodology tolerates higher errors in regions where there is more uncertainty about the potential power and enforces stricter error thresholds in regions where the potential power is more certain. This adaptive error tolerance is crucial for accurately identifying abnormal behavior without generating excessive false positives, particularly in regions where power output uncertainty is naturally higher. Thus, the proposed NBM methodology ensures more precise and context-aware detection of anomalies in wind turbine performance.

To validate the NBM methodology, we applied our model to detect anomalous observations within a dataset covering a two-month period. The model's predictions were subsequently compared against curtailments, shutdowns, and warnings documented in the status logs for the same timeframe. The results are summarized in Figure 4, which presents a confusion matrix comparing the events recorded in the status logs with the anomalies detected by the NBM methodology. In this matrix, True Positives (TP) represent correctly predicted abnormal events; True Negatives (TN) denote correctly predicted normal observa-
tions; False Positives (FP) indicate events flagged as abnormal by the model, though no corresponding power loss event was recorded in the status logs; and False Negatives (FN) are abnormal events that the model failed to detect.

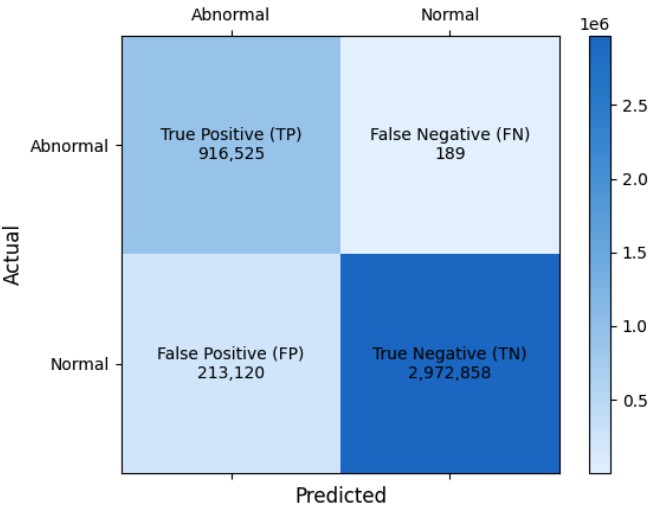

**Figure 4.** Confusion Matrix for NBM Methodology. Each entry represents the number of correctly predicted timestamps during the evaluation period.

    The majority of the data points in this table fall under True Positives or True Negatives, which are the desired outcomes of the methodology. Notably, the low number of False Negatives indicates that nearly all known abnormal events were successfully detected by the model. The few False Negatives that remain are turbine shutdowns in extremely low wind scenarios, where the
potential power is predicted to be almost zero. In these cases, the prediction error does not exceed the threshold for identifying abnormal behavior. The remaining data points are classified as False Positives, where the model identified anomalies without confirming evidence in the status logs.



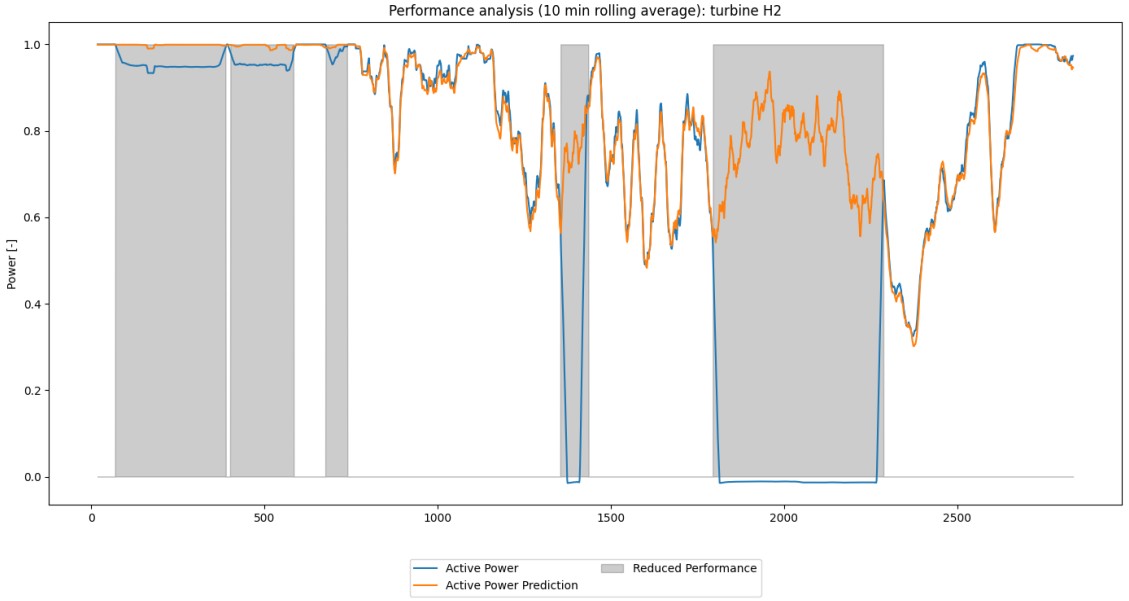

**Figure 5.** Performance analysis case study. A time series of the turbine's active power and the Spatio-Temporal GNN's prediction, smoothed using a 10-minute rolling average. The grey zones indicate abnormal performance events and are annotated using the NBM.

To further demonstrate the efficacy of our methodology in detecting power loss events, we conducted a detailed case study. Figure 5 illustrates a 24-hour period during which the active power output of a wind turbine, along with the corresponding predictions by the Spatio-Temporal GNN model, were analyzed. During this period, five loss events were detected, as indicated by the grey shaded areas. Since these events were confirmed by the status logs, they were categorized as True Positives.

The first three power loss events exhibit similar characteristics, as do the last two. In the first three events, the turbine's active power remains around 95% of its rated capacity, whereas the prediction model suggests that the turbine should be operating at full capacity. These power loss events are briefly interrupted by periods where the active power returns to the rated level. This pattern is consistent with a known curtailment strategy employed by the wind farm, where high wind speeds generate more power than can be transmitted to the grid, necessitating partial curtailment of the farm's output.

Figure 6 provides a broader perspective by aggregating data from individual turbines to represent the total active power output and the Spatio-Temporal GNN model's predictions for the entire wind farm. Additionally, the figure displays the number of turbines experiencing power loss events as detected by the NBM method.

During the period encompassing the first three power loss events, the total active power of the farm remains at its maximum capacity, while the predicted power exceeds this level. During this time, the farm controller actively manages the total output by curtailing certain turbines to avoid exceeding grid capacity limits.

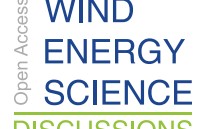

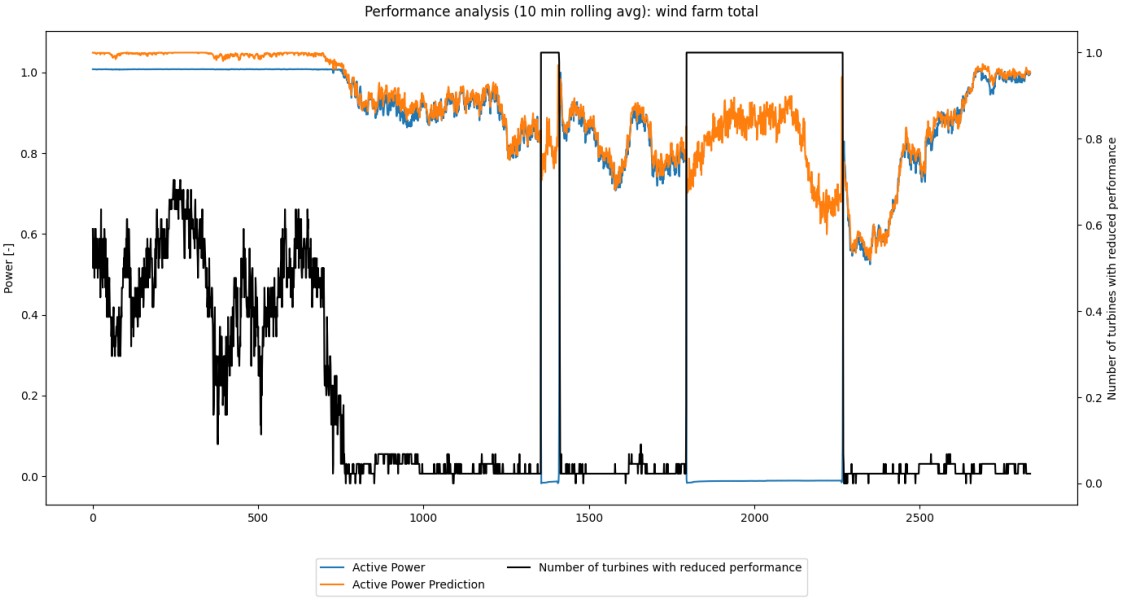

**Figure 6.** Performance analysis case study. A time series of the wind farm's total active power and the Spatio-Temporal GNN's prediction, smoothed using a 10-minute rolling average. The black line denotes the number of turbines with reduced performance, as detected by the NBM.

The last two events follow a different pattern, where the turbine shuts down completely while the prediction model continues to forecast nonzero active power. These significant prediction errors enable the NBM method to effectively detect these power loss events. Examination of the total wind farm data reveals that these shutdowns affected the entire farm, not just individual turbines. While a detailed root cause analysis is beyond the scope of this study, these events demonstrate the utility of our methods in identifying and analyzing power loss events. Furthermore, the ability to predict potential power output during these events provides valuable insights into the magnitude of power losses and their associated revenue impacts.

## 4.3 Analysis of unknown power loss events

It is worth noting that there is a considerable number of False Positives. This outcome is partially expected, given the lack of an exhaustive list of status logs containing all power loss events. However, upon closer examination, two wind turbines stand out for having a significantly higher number of False Positives compared to the rest of the wind farm.

The first case involves a wind turbine that consistently produces around 80% of its rated power, as shown in Figure 7. A brief analysis indicates that this is a deration of an individual turbine within the farm, a detail not captured in the available status logs. Since derations often result from spontaneous operator decisions and are typically not well-documented in status logs, automated post hoc detection of these events is valuable for ensuring accurate availability data. This enables operators to

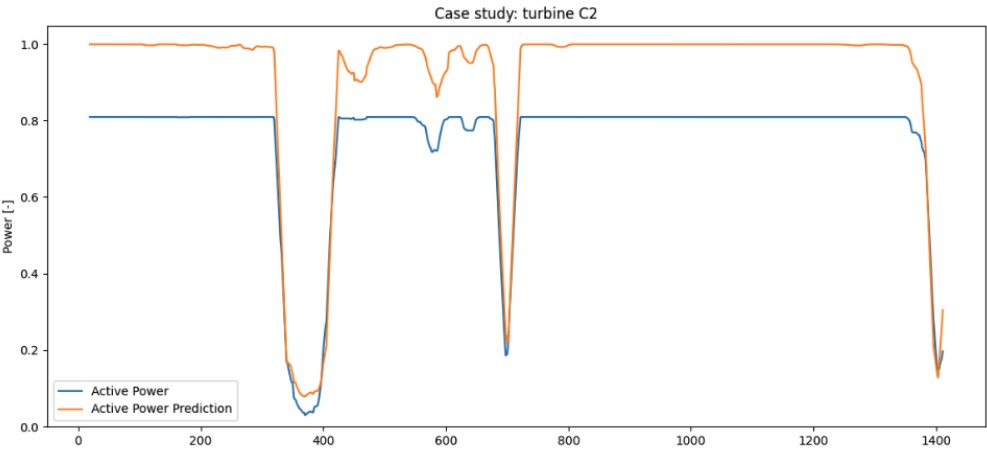

**Figure 7.** Performance analysis case study. A time series of the turbine's active power and the Spatio-Temporal GNN's prediction, both smoothed with a 10-minute rolling average, over a two-day period.

more precisely assess key performance indicators like turbine availability, quantify revenue losses due to derated turbines, and verify compliance with required performance standards.

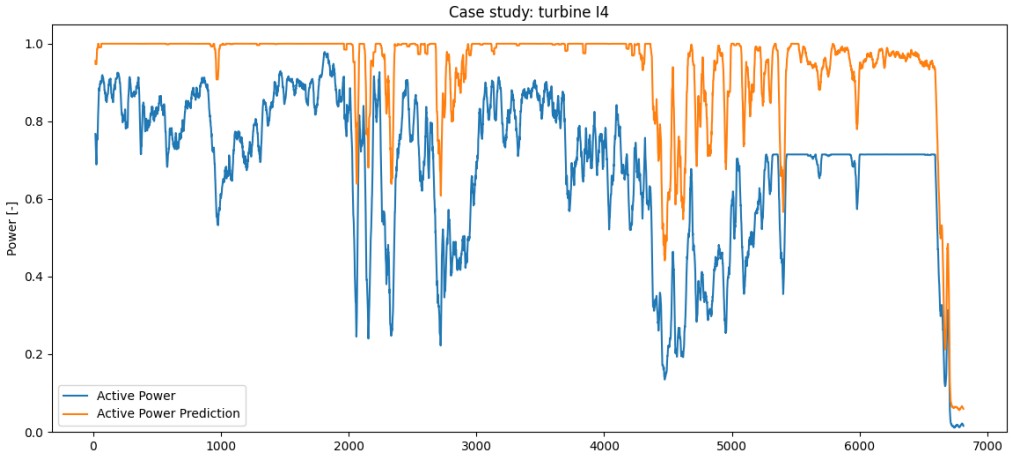

**Figure 8.** Performance analysis case study. A time series of the turbine's active power and the Spatio-Temporal GNN's prediction, both smoothed with a 10-minute rolling average, over a five-day period.

The second case is more complex and is depicted in Figure 8. In the days leading up to a maintenance event, we observed
large and continuous discrepancies between the turbine's active power output and the model's predictions. The turbine consistently produced less energy than expected, followed by a deration and eventual shutdown. Upon investigating this previously





unknown power loss event through historical logs, we discovered issues with the wind sensors on that turbine. These issues led to inconsistent pitch behavior, affecting the power output, as pitch regulation is dependent on accurate wind speed measurements from the sensors.

## 5 Conclusions

Our study introduces a robust and innovative approach to predicting the potential power output of wind turbines within a wind farm by leveraging the spatial and temporal dynamics of the wind environment. By modeling the wind farm as a graph and using Graph Neural Networks (GNNs) to aggregate information from neighboring turbines, we significantly improve the prediction accuracy compared to traditional power curve methods. The addition of a temporal component through a hybrid GNN-LSTM model further enhances the model's ability to capture and leverage temporal patterns in the data, providing superior performance in terms of Mean Absolute Error (MAE) and Mean Absolute Percentage Error (MAPE).

The integration of our prediction model into a Normal Behavior Model (NBM) framework shows its practical utility in analyzing turbine underperformance and identifying power loss events. Our model's ability to detect abnormal behavior with high accuracy, even in the presence of complex wake interactions and varying wind conditions, demonstrates its potential for real-world applications. The case studies presented validate the model's effectiveness in detecting grid curtailments, shutdowns, individual turbine derations, and anomalous behavior, offering valuable insights into the operational performance of wind farms.

Future work could aim to enhance model accuracy by incorporating additional environmental factors and expanding its application to different wind farms. Additionally, we plan to explore the use of this power prediction method in wind farm control. In a control setting, the potential power output of each turbine is crucial for determining optimal power setpoints. However, as the selected power setpoints and intra-farm wakes influence turbine loads, it will also be necessary to model the load spectrum present in wind farms (Verstraeten et al., 2019; Nejad et al., 2022).

*Author contributions.* SD: Conceptualization, Formal analysis, Investigation, Methodology, Writing – original draft preparation, & editing; TV: Conceptualization, Supervision, Writing – review & editing; PD: Formal analysis; AN: Funding acquisition, Supervision; JH: Conceptualization, Funding acquisition, Supervision.

*Competing interests.* The authors declare that they have no conflict of interest.

*Acknowledgements.* The authors would like to acknowledge the Energy Transition Funds for their support through the POSEIDON and BeFORECAST projects. This research was supported by funding from the Flemish Government under the "Onderzoeksprogramma Artificiële Intelligentie (AI) Vlaanderen" program.





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
