# Peer review of "Spatio-Temporal Graph Neural Networks for Power Prediction in Offshore Wind Farms Using SCADA Data"

_Wind Energy Science, 2024_

## Referee Comment (RC2)

[referee-annotated manuscript omitted]

---

## Author Response (AR1)

**Reviewer 1**

We thank the reviewer for their constructive feedback. We provide our answers to the comments below, structured in the following sequence: (1) comments from referees/public, (2) author's response, and (3) author's changes in the manuscript.

- 1) Pg. 2, ln. 53: Please define LSTM.
  2) We added a definition in the abstract (when the word is first introduced).
  3) "A temporal component is introduced by feeding a time series of input features into the graph, processed through a Long Short-Term Memory (LSTM) network before being passed to the GNN."
- 1) Pg. 3, In. 69: "it cannot provide a precise forecast of wind power as it fails to accurately model the nonlinear relationship between wind speed and power output": A power curve does not necessarily have to be a linear function, and can capture the nonlinear relationship between wind speed and power if modeled correctly. Can you explain this statement in more detail? 2) Indeed, it can capture the nonlinear relationship between wind speed and power, however, it has other limitations. For example, it does not account for inter-turbine interactions and temporal patterns; and it does not consider other environmental variables besides wind speed, which could positively affect the prediction accuracy. We rephrased the sentence to clarify this. 3) "Traditionally, the manufacturer's power curve, which describes the theoretical behavior of a wind turbine in constant and low turbulence wind conditions, is used to predict the expected power output at different wind speeds. However, this method fails to account for the specific atmospheric conditions in a particular wind farm. Therefore, most wind farm operators use the method of binning (International Electrotechnical Commission (IEC), 2017) to estimate the power curve based on measurement data from their own wind turbines. With this method, the range of measured wind speeds is partitioned into separate bins of 0.5 m/s, and the power response is calculated by averaging the power data falling in each bin. While this method is easy to implement and can capture the primary nonlinear relationship between wind speed and power output, it does not account for inter-turbine interactions and varying environmental conditions such as turbulence intensity."
- 1) Pg. 5, ln. 127: "both computed over a 10-minute interval centered around the 1-second data point.": Does this mean that the method for predicting power outputs will have at least a 5-minute lag, because the computed TI value is delayed by 5 minutes?

2) Yes, indeed, the current TI calculation method introduces a 5-minute lag for real-time applications. However, we currently don't use this method for real-time predictions, so in our case this is not a problem. We clarified this in the manuscript.

3) "both computed over a 10-minute interval centered around the 1-second data point. In practical terms, this method requires wind speed data from 5 minutes before and 5 minutes after each point to compute the turbulence intensity. As a result, any real-time application of this calculation would inherently introduce a 5-minute lag, since the full 10-minute window is not complete until 5 minutes after the moment being analyzed. However, in our case, this method is used for historical data analysis rather than real-time predictions, so this lag does not impact the accuracy or usability of our power predictions."

• 1) Pg. 5, ln. 137: "uses properties of the power curve based on the IEC standard to annotate steady-state control conditions.": Please mention which IEC standard is used and provide a reference.

2) We explained the filtering approach in more detail and included a reference to the IEC standard.

3) "To achieve this, we used a physics-based filtration method, which relies on the properties of the power curve as per IEC standards (International Electrotechnical Commission (IEC), 2017) to annotate steady-state control conditions. Using this approach, we classified the data into different operating regions (i.e., torque control, pitch control), flagged data points falling outside these regions as abnormal, and removed them from the dataset."

 1) Pg. 5, ln. 138: "Using this approach, different control regimes can be identified, and outliers and abnormal behavior can be removed from the dataset.": Do you remove entire timestamps (for all turbines) when any individual turbine has an outlier/abnormal operation identified? Or do you only remove the data for the specific turbine with abnormal operation?

Pg. 5, ln. 140: "lagged values of the input features are incorporated into the model": How many samples in the past are used? Do you just use the current and previous timestamps, or several timestamps in the past?

Pg. 5, ln. 142: "to ensure that the lagged values utilized by the model are consecutive and there are no discontinuities due to missing data.": Can you explain the criteria for determining continuous data? Are you identifying timestamps where there is no missing data for all turbines for the current and lagged timestamps? Further, for a large wind farm, it could be rare that all turbines are operating normally at any given time (due to repairs, derating, etc.). How do you handle the case when there are always some turbines with missing data?

Pg. 5, ln. 143: "partitioned into distinct subsets for model training, validation, and testing": A brief explanation of these three subsets would be helpful for readers not as familiar with AI/ML training and validation.

2) We have rewritten the paragraph concerning these comments and will sum up the main changes and clarifications.

- We remove the entire timestamp when any turbine has abnormal operation identified. Indeed, this leaves us with fewer datapoints for model training, but we found that this approach works best for creating a normal behavior model.
- We clarified that we used lagged values for 5 minutes in the past.
- Our criterion for continuous data requires that no data is missing across all turbines for the current and lagged timestamps. This leaves us with less data, but for the farm we considered it seemed to be no issue. We will remember this for future work in case we want to develop a similar model for a wind farm where this poses a problem. In that case, we could look at dynamically changing the graph based on which turbines are operating normally and thus only discarding data from abnormally operating turbines.
- We explained the purpose of train/validation/test datasets.

3) "We removed the entire timestamp across all turbines when any individual turbine exhibited abnormal operation. While this reduces the number of data points available for model training, we found that this approach yields the best results in creating a reliable NBM. To leverage temporal patterns in the wind flow throughout the wind farm, we incorporated lagged values of the input features into the model. Specifically, a time series of input features between the prediction time t and t – T (with T = 5 minutes) was included. Given the time-series nature of the data, it is essential to maintain temporal continuity, so a final filtering step was implemented to ensure that lagged values used by the model are consecutive and free from discontinuities caused by missing data. To ensure continuity in the time series, we required that no timestamps were missing across all turbines for the current and lagged timestamps. Finally, the resulting dataset was partitioned into distinct subsets for model training, validation, and testing. The first year of data was used as training data and was used to fit the model parameters. The second year of data was split into equal parts as the validation and test set. Validation data is utilized to tune hyperparameters and prevent overfitting, and testing data is used to assess the model's performance on unseen data, ensuring its generalizability."

- 1) Algorithm 1: Please mention the value of "tolerance" that is used.
  2) We mentioned the value of the chosen tolerance.
  3) "if |d1 + d2-d\_direct|< tolerance (250m) then"</li>
- 1) Pg. 7, In. 154: "containing a set of geometric features, such as the length and direction of each edge": Can you state exactly which features are used as edge attributes in this work? This sentence makes it sounds like length and direction are just two examples of what could be used.
  2) We changed our wording to 'specifically' to make it clear only length and direction are used.
  3) "Similarly, a vector of edge attributes can be defined for each edge, containing a set of geometric features, specifically the length and direction of each edge."

1) Pg. 7, Ins. 175-183: These two paragraphs contain a lot of terms that require further explanation for readers less familiar with neural networks. For example, "generalized message aggregators", "pre-activation residual connectors", "message normalization layers", "feature encoder", "sigmoid activation", "multi-layer perceptrons", "ReLU", "DeepGCNLayer", "preactivation residual connections". I would suggest describing these terms in more detail and what their purpose is in the GNN model. Or references could be provided for some of the terms. Also, in addition to Figure 1, a block diagram outlining the entire GNN model from the inputs to the estimated power outputs would greatly clarify the model and put some of these terms in context.

Pg. 8, lns. 184-186: Similarly, please explain "LSTM" and "node feature encoder" and how they are used in the model. A figure showing the full model architecture for the Spatio-Temporal GNN would be helpful as well to clarify how it differs from the spatial GNN.

Figure 1: Please label the subfigures in the figure (a, b, c, d).

Figure 1: Please describe what specific functions are used for the "UPDATE" and "AGGREGATE" functions. Earlier in Section 3.3 you mention that they are arbitrary differentiable function (i.e., neural networks), but is isn't clear what is actually used in this work. Further, the equations in Fig. 1c do not match the form of the equation in line 169. Is one of these incorrect? If so, please make sure they match or clarify which form is actually implemented in your work.

2) We have rewritten these paragraphs and removed the unnecessary terms for more clarity. Also, we adapted the figure to capture the entire Spatial and Spatio-Temporal GNN model.

- We removed the confusing terms since they are not required to understand the model architecture. Interested readers can find explanations for these in the GENConv paper by Li et al. (2020).
- We adapted the figure to capture the entire Spatial and Spatio-Temporal GNN model, from inputs to the feature encoders (c and d), then message passing (e), and finally the decoder to make the final predictions(f).
- We labeled the subfigures.
- In the previous version, we explained the concept of general GNNs with their general equations, where UPDATE and AGGREGATE are arbitrary differentiable functions. Since we us a specific GNN model (GENConv) in this work, we changed the functions to the specific message passing function for GENConv. We also explained which UPDATE and AGGREGATE functions are used in this work (a multilayer perceptron (MLP) as update function and the softmax aggregation)
- The functions displayed in the equation and the figure now match and correspond to the specific message passing functions used in GENConv.

3) See Section 3.3.

1) Pg. 8, In. 194: "based on the objective function.": What specific objective function is used? Is it the MSE of the turbine power for all turbines?
2) It is indeed the MSE between the turbine power and the predicted power for all turbines. We rephrased the sentence to make it clearer.

3) "Optuna automatically searched for the optimal hyperparameters by minimizing the objective function, defined as the Mean Squared Error (MSE) between the model's predictions and the target values."

- 1) Pg. 8, ln. 196: "Hyperparameter combinations that resulted in the lowest MSE for the validation dataset were retained...": What were the specific combinations that were ultimately selected for the final model?
  2) We added this to the table.
  3) n/a
- 1) Table 3: Please explain the hyperparameters and how they are used in the GNN models.
  2) We added a description of the hyperparameters to the table.
  3) n/a

- 1) Section 4.1: Many of the conclusions of this work are based on a comparison of the GNN performance with the simple power curve approach. Please discuss in more detail what kind of power curve model is used. This will help put the results in context.
  2) We provided a more thorough explanation of the power curve method earlier in the paper , and clarified that we used the power curve binning method from IEC 61400-12.
  3) Line 74-77: Therefore, most wind farm operators use the method of binning (International Electrotechnical Commission (IEC), 2017) to estimate the power curve based on measurement data from their own wind turbines. With this method, the range of measured wind speeds is partitioned into separate bins of 0.5 m/s, and the power response is calculated by averaging the power data falling in each bin."
  Line 234-235: "The models are compared with a Power Curve model based on the power curve binning method (International Electrotechnical Commission 2.5 m/s) and the power Curve model based on the power curve binning method (International Electrotechnical Commission 2.5 m/s) as a baseline."
- 1) Pg. 12, ln. 237: "computed for each wind direction bin": How do you estimate the wind direction?
  2) We used the wind direction measurements of the turbine that is positioned upstream with respect to the dominant wind direction as measured by its wind vane.
  3) n/a
- 1) Pg. 12, ln. 241: "One turbine is positioned in the free flow relative to the dominant direction": Is this the turbine in the numerator or denominator of Eq. 2?
  2) This is the turbine in the numerator of Eq. 2. We also clarified this in the text.
  3) "One turbine is positioned in the free flow relative to the dominant wind direction, while the other is located in a subsequent row behind the first turbine, thus experiencing the wake effect generated by the upstream turbine (Turbine 1 and Turbine 2, respectively, as indicated in Eq. (2))."
- 1) Pg. 12, ln. 245: "Our spatio-temporal GNN model demonstrates remarkable agreement with the energy ratios from the SCADA data": Can you include the power curve-based energy ratio estimates in Fig. 3? This would help show how much value there is in using the GNN model instead of a simple power curve estimator.
  2) We included the power curve method in Fig.3, and we quantified the differences in performance between both methods by calculating the wind-direction-frequency-weighted average of the differences in predicted vs actual energy ratios.
  3) n/a
- 1) Pg. 12, In. 255: "shows the Spatio-Temporal GNN model's effectiveness in capturing complex dynamics of wake interactions within the wind farm.": Much of the complex wake behavior would be captured by the turbines' own wind speed measurements, which are used as features in the GNN model and would also be the inputs to the simple power curve model. So it seems likely that the simple power curve model would already capture these wake dynamics.
  2) It is indeed true that some of the wake behavior is captured in a turbine's own wind speed measurements. However, these measurements can be heavily subjected to noise and are therefore not always accurate. With our GNN approach, instead of relying on only one wind speed measurement (the one of the turbine itself), we aggregate multiple measurements into a prediction. Additionally, since the GNN model learns a relationship between different turbines' wind measurements, their relative position with respect to each other, and the power output, the wake interactions are learned by the model. To validate this, we added the power curve to the comparison for the energy ratio plot, and show that especially in wake-affected wind directions, our GNN proves more accurate.
- 1) Sections 4.2 and 4.3: For the results in these sections, please mention whether you are using the spatial GNN or the spatio-temporal GNN.
  2) We added it to the text in section 4.2, as well as in the captions of all relevant figures.
  3) n/a

- 1) Pg. 12, In. 264: "the standard deviation of the produced power for a given wind speed": Should this be "for a given wind speed bin"? And if so, what bin width is used?
  2) Indeed, this should be for a given wind speed bin. The bin width we used was 0.5 m/s. We added this to the manuscript.
  3) "In our methodology, anomalies are flagged when the deviation between the predicted and actual power exceeds two times the standard deviation of the produced power for a given wind speed bin."
- 1) Fig. 4: Comparing these results to the confusion matrix results when using the simple power curve model would show how much value there is in using the GNN model. If the results are similar, it seems like it would be preferable to just use the simple power curve model.
  2) We added the power curve to the comparison and showed that the GNN can obtain better results.
  3) n/a
- 1) Figs. 5 and 6: Can you also compare these results to the estimated reduced performance periods based on the simple power curve model?
  2) We quantitatively compared the results of the GNN with the power curve in the confusion matrix.
  3) n/a
- 1) Fig. 6: The "number of turbines with reduced performance" axis label should include "normalized" since the values are between 0 and 1.
  2) We added this to the axis label.
  3) n/a

**Reviewer 2**

We thank the reviewer for their constructive feedback. We provide our answers to the comments below, structured in the following sequence: (1) comments from referees/public, (2) author's response, and (3) author's changes in the manuscript.

• 1) Line 4: Define

2) We defined 'LSTM' before introducing the word.
3) "A temporal component is introduced by feeding a time series of input features into the graph, processed through a Long Short-Term Memory (LSTM) network before being passed to the GNN."

**• 1) Line 6-10: Quantify this**

2) We quantified the relative improvements in MAE and MAPE, as well as the improvements in capturing abnormal events of our models compared to the power curve binning method.
3) "The results show that both the Spatial and Spatio-Temporal GNN models outperform traditional data-driven power curve methods, achieving reductions in Mean Absolute Error (MAE) of approximately 22.6% and 30.3%, respectively, and in Mean Absolute Percentage Error (MAPE) of around 20.7% and 30.5%."

"Additionally, the model achieves remarkable agreement with SCADA-derived energy ratios across the full range of wind directions, with a weighted average error of 0.0373; an improvement of approximately 57.4% compared to the power curve binning method."

" Compared to the power curve method, the Spatio-Temporal GNN reduces the rate of undetected power loss events from 12.6% to just 0.02%, demonstrating a substantial improvement in capturing abnormal events."

• 1) Line 12-20: Add some references

2) We added some references.

3) "Wind energy plays a vital role in addressing the escalating global energy demand while aligning with sustainability goals to combat climate change. As countries worldwide strive to reduce greenhouse gas emissions and transition towards renewable energy sources, wind energy emerges as a clean and abundant resource capable of meeting a significant portion of our energy needs (Yousefi et al., 2019). Offshore wind farms offer promising opportunities to harness stronger and more consistent wind speeds compared to onshore locations, thereby contributing to a more resilient and sustainable energy infrastructure (International Energy Agency, 2023). As technology advances and costs continue to decline, offshore wind projects are becoming more economically viable and attractive investments for governments, developers, and energy consumers alike (European Commission, 2023)."

• 1) Line 23: which is important because...? (please add)

2) We changed the sentence to clarify that choosing operating settings and maintenance schedules based on a turbine's potential power can help maximize production and minimize cost.

3) "By understanding the potential power each turbine could generate under different wind conditions, operators can make informed decisions about operation settings and maintenance schedules to maximize production while minimizing costs."

 1) Line 25-27: Add a review of existing literature for power predictions & add a sentence about why existing methods are not accurate enough. Line 39: What are current methods and why aren't they good enough? Line43: Add a review of existing literature, including but not limited to previous applications of GNNs for wake effects such as https://iopscience.iop.org/article/10.1088/1742-6596/2647/11/112006 and https://iopscience.iop.org/article/10.1088/1742-6596/2505/1/012047/pdf, but also other hybrid and data-driven methods. The literature reviews in the above-mentioned papers might also be helpful. Line 57: Some of my previous comments have been (partly) answered in this section. I would suggest combining Section 1 and Section 2, introducing the problem, the existing solutions / previous work, and the gap your work is intending to fill.

2) We added the suggested papers to the literature review, and clarified in the introduction how our method is different from existing literature. We decided to clarify how our work differs from the overall literature as part of the motivation of our work. We kept a separate section of the more detailed description for related work in order to maintain focus on describing the general scope of the paper in the introduction. For more clarity about the structure of the paper and to show the readers our intention to discuss related work, we added an outline summarizing the structure of the paper at the end of the introduction.

3) "... Finally, the development of state-of-the-art wind farm control methods typically relies on models that capture the wake effect between wind turbines (Verstraeten et al., 2021). Wake effects are highly non-linear and difficult to capture, and traditionally modeled using physics-based wake models that can be calibrated using real-world data (Van Binsbergen et al., 2024a, b, c). In this work, we propose a model to predict the potential power of wind turbines operating in wakes, specifically designed for wind farm control applications. The model utilizes integrated measurements from the supervisory control and data acquisition (SCADA) system of multiple turbines at a high temporal resolution of 30 seconds. The remainder of this paper is structured as follows: In Sect. 2, we position our research within the context of existing literature. Sect. 3 details our approach, including the data processing pipeline, the GNN framework, and the training and hyper-parameter tuning process of our models. Sect. 4 presents a comprehensive analysis of the model's performance, including its predictive accuracy and ability to detect abnormal events, compared against baseline methods. Finally, in Sect. 6, we summarize the key findings of our study and outline potential directions for further research."

- 1) Line 29: It is not clear what this means. Please clarify.
  2) We changed our wording to make clear which temporal and spatial patterns are meant.
  3) "Can we leverage temporal trends in wind flow, turbine positioning, and integrated measurements from different turbines within the wind farm to improve power prediction accuracy?"
- 1) Line 39: Do you mean "quantifying"? Whether you mean "quantifying" or "understanding", you should explain why and how quantifying or understanding these losses can lead to the mentioned improvements.

2) We changed the wording to 'quantifying' and elaborated on the importance of quantifying losses during curtailments.

3) "By quantifying the magnitude and frequency of power losses during curtailments, operators can identify patterns and evaluate the economic impact of curtailments."

**• 1) Line 43: Quantify this**

2) We left out 'high temporal resolution' to maintain focus on the core aspect of the mentioned paper, which is modeling wake effects.

3) "Finally, the development of state-of-the-art wind farm control methods typically relies on models that capture the wake effect between wind turbines (Verstraeten et al., 2021)."

• 1) Line 44-45: Your added value compared to these studies would be the higher temporal resolution

2) We added our added value of higher temporal resolution, which is specifically beneficial for control applications.

3) "In this work, we propose a model to predict the potential power of wind turbines operating in wakes, specifically designed for wind farm control applications. The model utilizes integrated measurements from the supervisory control and data acquisition (SCADA) system of multiple turbines at a high temporal resolution of 30 seconds."

• 1) Line 48-56: I would recommend moving the content of these bullet points to the conclusions. It is too early in the paper to fully understand what is meant by many of these things. Instead, you could say what the overall goal is and then summarise the structure of the paper.

2) We left out this part since the content was already present in the conclusion section. We also added an outline summarizing the structure of the paper.

3) "The remainder of this paper is structured as follows: In Sect. 2, we position our research within the context of existing literature. Sect. 3 details our approach, including the data processing pipeline, the GNN framework, and the training and hyper-parameter tuning process of our models. Sect. 4 presents a comprehensive analysis of the model's performance, including its predictive accuracy and ability to detect abnormal events, compared against baseline methods. Finally, in Sect. 6, we summarize the key findings of our study and outline potential directions for further research."

• 1) Line 58-59: Please add what the application is (for power predictions of wind turbines using SCADA data?) & Quantify this

2) The application of the NBM by Lyons and Gocmen (2021) was added, and we clarified that the effectiveness of the methodology was shown by qualitative analysis of instances of over- and underperformance.

3) "An example of a normal behavior model for performance analysis, based on an artificial neural network trained on an abnormality-filtered SCADA dataset for power prediction, is shown by Lyons and Gocmen (2021). The authors demonstrate the effectiveness of the developed NBM for power performance analysis by qualitatively discussing instances of over- and underperformance identified by the model."

• 1) Line 62-63: For which application? Line 65: Quantify?

2) We added that the model is used for condition monitoring of wind turbines and that this study evaluates their NBM by qualitative analysis of a variety of detected faults.

3) "In another study by Bilendo et al. (2022), a different normal behavior model is introduced for condition monitoring of wind turbines, leveraging a heterogeneous stacked regressor (HET-SR) algorithm. This algorithm learns from optimal power curve data to serve as a predictive model within their NBM framework. While a qualitative analysis of a variety of faults that can be detected by the model is shown, employing more advanced prediction models that account for additional variables beyond wind speed could offer opportunities for a more comprehensive performance analysis beyond general fault detection."

1) Line 68: I don't think you mean that. Either the manufacturer's power curve is traditionally used to predict performance, or the power curve describes the theoretical behaviour of the wind turbine in constant and low turbulence wind conditions. Please rephrase!
Line 69: I would suggest a proper description of what the power curve is, rather than just "a simple relationship", given the importance of the topic for this paper.
2) This section was rephrased to properly introduce the power curve and the binning method.
3) "Therefore, most wind farm operators use the method of binning (International Electrotechnical Commission (IEC), 2017) to estimate the power curve based on measurement data from their own wind turbines. With this method, the range of measured wind speeds is partitioned into separate bins of 0.5 m/s, and the power response is calculated by averaging the power data falling in each bin."

- 1) Line 69: Also it fails to predict the wind speed. Either add a comment about that or clarify that wind speed prediction is not the topic of this paper.
  2) We clarified in the introduction that wind speed prediction is not the topic of this paper by immediately emphasizing our method is based on SCADA data.
  3) "The model utilizes integrated measurements from the supervisory control and data acquisition (SCADA) system of multiple turbines at a high temporal resolution of 30 seconds."
- 1) Line 92: Define this

2) We clarified 'LSTM' in the abstract.

3) "A temporal component is introduced by feeding a time series of input features into the graph, processed through a Long Short-Term Memory (LSTM) network before being passed to the GNN."

1) Line 109-110: You mention Section 3.1 and Section 3.3, but not Section 3.2 or Section 3.4. This seems strange. Please include all four sub-sections in this introduction.
2) We thank the reviewer for noticing this oversight. We included all four subsections in the introduction.
3) "The data as well as the preprocessing pipeline are described in detail in Sect. 3.1. In Sect. 3.2,

3) "The data as well as the preprocessing pipeline are described in detail in Sect. 3.1. In Sect. 3.2, the graph representation of the wind farm is introduced. The different model architectures and characteristics are elaborated on in Sect. 3.3, and finally, the hyperparameter tuning process is explained in Sect. 3.4."

- 1) Line 111: Please check for consistency in tense. You switch past and present tense. Please be consistent. If you are talking about something that was done in this work, you should use past tense. If you are talking about how a method works, you should use present tense.
  2) We checked for consistency in tense and made changes where necessary.
  3) n/a
- 1) Line 113: Define the first time you use it only
  2) We removed the definition here and defined it the first time we use it in the paper.
  3) n/a
- 1) Line 116-117: Mention how to deal with different naming conventions between wind turbine types.

2) This is indeed something that differs per turbine and will always be an issue. We added a paragraph to future work, since this will be important when we extend our methodology to multiple wind farms. For this paper, since we only apply the model to one wind farm, we opted to keep the focus of the paper on the model rather than about data standardization. For more clarity, we rephrased the sentence as well.

3) "As most contemporary wind farms have the used input signals readily available (e.g., through their SCADA system), the model is easily accessible and applicable to other wind farms." & "Developing a model that operates effectively across multiple wind farms presents a promising research direction. Achieving this, however, requires further efforts in data standardization and the consolidation of varying data formats across wind farms. In this context, the use of ontologies and taxonomies, such as RDSPP, could facilitate consistent data integration."

• 1) Line 119: Please explain why these features were selected and why other possible features were not or could not be used.

2) We added a sentence explaining that we chose these input features because they directly capture the key physical factors influencing wind farm power generation, such as wind resource characteristics, directional influences, and flow variability. Also, we mentioned that other features were excluded due to data availability limitations, potential redundancy, and a focus on simplifying the model while retaining interpretability and accuracy.

3) "The input features to the prediction model are wind\_speed, wind\_direction\_sin, wind\_direction\_cos, and turbulence\_intensity. They were chosen because they directly capture the key physical factors influencing wind farm power generation, such as wind resource characteristics, directional influences, and flow variability. Other features were excluded due to data availability limitations, potential redundancy, and a focus on simplifying the model while retaining interpretability and accuracy."

1) Line 121: measured by a wind vane mounted on the nacelle? Clarify if you are talking about wind direction relative to the nacelle position or absolute wind direction, in which case this signal would be a subtraction of the wind vane and the nacelle position.
 2) We elerified that we used the absolute wind direction and that it is measured by a wind vane

2) We clarified that we used the absolute wind direction and that it is measured by a wind vane located on the nacelle.

3) "The absolute wind direction, measured by a wind vane mounted on the nacelle, was represented by sine and cosine transformations to account for its circular nature."

- 1) Table 1: I suggest adding units to this table & The pitch angle is not the same thing as the angle of attack. Please clarify.
  2) We added units to the table and changed the definition of pitch angle.
  3) "The angle between the plane of rotation and the chord line of the blade [°]"
- 1) Line 133-134: Please elaborate
  2) We elaborated on the fact that with this resolution, computational costs remain manageable, fluctuations are smoothed, and key temporal dynamics are preserved.
  3) "First, the raw SCADA data with a sampling frequency of 1 Hz was resampled to 30-second averages as a trade-off between high temporal resolution, acceptable noise levels, and manageable computational costs. This approach smooths out rapid fluctuations while preserving key temporal dynamics."
- 1) Line 137-139: *This approach should be explained in more detail.* & *Include a reference* 2) We explained the filtering approach in more detail and included a reference to the IEC standard.

3) "To achieve this, we used a physics-based filtration method, which relies on the properties of the power curve as per IEC standards (International Electrotechnical Commission (IEC), 2017) to annotate steady-state control conditions. Using this approach, we classified the data into different operating regions (i.e., torque control, pitch control), flagged data points falling outside these regions as abnormal, and removed them from the dataset."

• 1) Line 139-140: Please explain this more thoroughly. Does it mean that you use input data from previous periods as an input for a current time period? How far is the lag (and why), and which features were used?

2) We clarified in our explanation that we use lagged values of the input features as input for the model. We also added that we use a lag of 5 minutes.

3) "To leverage temporal patterns in the wind flow throughout the wind farm, we incorporated lagged values of the input features into the model. Specifically, a time series of input features between the prediction time t and t – T (with T = 5 minutes) was included."

• 1) Line 143: Based on what?

2) We added an explanation in the manuscript on how we split the data and for what purpose each dataset was used.

3) "Finally, the resulting dataset was partitioned into distinct subsets for model training, validation, and testing. The first year of data was used as training data and was used to fit the model parameters. The second year of data was split into equal parts as the validation and test set. Validation data is utilized to tune hyperparameters and prevent overfitting, and testing data is used to assess the model's performance on unseen data, ensuring its generalizability."

- 1) Line 157-160: This text fits in the literature review, not here.
  2) We moved this part to the literature review.
  3) n/a
- 1) Line 174: Why this model?
  2) We clarified that we used this model because of its ability to incorporate edge features in the message passing process.
  3) "Both models utilize the GENeralized Graph Convolution (GENConv) model proposed by Li et al. (2020) due to its ability to incorporate edge features into the message-passing process."

- 1) Line 184: *Explain this*2) We defined LSTM earlier in the paper.
  3) n/a
- 1) Figure 1: *The labels (a) to (d) are missing*2) We added the labels to the figure.
  3) n/a
- 1) Line 189: maximising its transferability?
  2) We do in fact mean 'generalizes' (in the machine learning), not 'maximizing transferability', since we want to focus on the model's ability to perform effectively on unseen data within a similar context. In machine learning, 'transferability' might entail that we want our model to apply learned patterns to different contexts (such as unseen wind farms with different layouts or conditions).
  3) n/a
- 1) Line 193: Meaning the most optimal ranges established in the study? Please make this more clear.

2) We changed our wording to 'proposed' ranges for more clarity. We also added the optimized hyperparameters to the table and description.

3) "Table 3 lists the hyperparameters that were tuned, along with their proposed ranges, as well as the optimized value for both models."

• 1) Line 205: from the binned average in 1 m/s bins? (also, refer to the power curve binning method in IEC 61400-12)

2) We clarified the power curve method in section 2, together with a reference to the IEC standard.

3) "The models are compared with a Power Curve model based on the power curve binning method (International Electrotechnical Commission (IEC), 2017) as a baseline."

- 1) Line 206-207: Remove this. You already described the power curve earlier (at least you will do when you respond to my earlier comments :-).
  2) We removed this sentence and explained it earlier in the paper.
  3) n/a
- 1) Table 4: *To which value*?

2) We clarified that the values have been normalized by dividing by the rated power.3) "Performance Metrics. The MAE values have been normalized by dividing them by the turbine's rated power for confidentiality reasons. The lowest errors are highlighted to showcase the model with the best predictive performance on each dataset."

• 1) Line 209-210: Not really, it shows the advantage of not using the power curve method. There are plenty of other data-driven methods that show superior performance for power prediction compared to power curve binning, e.g. https://doi.org/10.1115/1.4053513 & It's difficult to compare different methods directly due to the different data sets used; however, I expect these results somehow to be put in relation to the general improvement expected from other data-driven models.

2) We moved this sentence to the discussion, after our argumentation of why GNNs perform better in our experiments. In addition, we compared our method with a standard MLP, a common data-driven model, and added evidence to the results section to make a stronger case for GNNs.
3) See Section 4 & 5.

1) Line 227-229: And compared to previous work with other models?
2) We compared our results, both for error metrics (MAE, MAPE) and energy ratio predictions with an MLP, showing how our model exhibits better performance.
3) n/a

1) Line 236-237: You should explicitly say that the sum of powers over a fixed amount of time is equivalent to the energy, and because a ratio is made over the same time period, the time drops out. Otherwise it's confusing that you are calling it energy ratio but adding powers.
 2) We clarified this in the manuscript.

3) "This sum of power measurements over a fixed period is equivalent to the total energy produced during that time. Because the ratio is calculated over the same time period for both the test and reference turbines, the time factor cancels out."

• 1) Line 245: Can you quantify the differences in energy ratio? Either plot the differences in energy ratio between model and SCADA and/or take the wind direction frequency weighted average of the differences?

2) We quantified the differences in energy ratio by calculating the wind-direction-frequencyweighted average of both the power curve method and the Spatio-Temporal GNN. We also plotted the differences in energy ratio between our model and SCADA and showed the advantages of using GNNs, especially in waked conditions.

3) "To quantify this, we calculated the wind-direction-frequency-weighted average of the differences between the predictions and the actual energy ratios. The power curve method has an average error of 0.0875, and the MLP achieves an average error of 0.0828. In contrast, the Spatio-Temporal GNN attains a significantly lower average error of 0.0373, representing a relative improvement of approximately 57.4% over the power curve binning method."

1) Line 249-257: But it's more important to consider how the difference between model and SCADA behaves with wind direction, surely? If you do the analysis suggested above, you can then examine how well the model works with wind direction. Does it work well in waked flow or not? Isn't the whole point of the GNN that it can predict wake effects well/better than other methods? & I don't think you have showed this, unless you do the further analysis suggested above.
2) We did the suggested analysis and showed that our model works well in waked flow (see Section 4.1).

3) "The energy ratio error plot reveals that the Spatio-Temporal GNN significantly outperforms both the power curve binning method and the MLP, particularly under waked conditions. This highlights the strength of our GNN in accurately predicting potential power in waked flows, where conventional methods like the power curve binning and MLP models fall short."

- 1) Line 258: I suggest a comparison between these results and the results using a different method, such as the power curve binning method, or another method from the literature
  2) We added a comparison with the power curve binning method for both the energy ratio part and the quantitative analysis of the confusion matrix.
  3) See Section 4.2.
- 1) Line 263-265: Why did you choose this?
  2) We clarified that we use a commonly used threshold-based methodology for anomaly

detection. Through experimentation, we found that 2\*std was a good threshold for flagging anomalies. We also clarified that the NBM is not our primary focus and therefore no further optimization was done.

3) "While our primary objective is not to develop a Normal Behavior Modeling (NBM) framework, we aim to validate the robustness of our power prediction model in scenarios where no ground truth is available, such as turbine shutdowns or curtailments. To achieve this, we employ a commonly used anomaly detection methodology: a threshold-based approach that identifies anomalies by analyzing the residuals between the predicted and actual values."

• 1) Line 266-267: I don't like this sentence. First of all, "the SD is low because the value is low" is not really an argument. Second, why is the power output more predictable when it is low? Line 270: It's not clear how you approach ensures this. Above you mention 3\*SD but not a variable approach. Please clarify.

2) We rephrased the paragraph to explain better how the standard deviation evolves with wind

speed and thus clarifying how our approach is variable based on the inherent uncertainty in produced power.

3) "At low wind speeds, the standard deviation of the produced power is relatively low due to the reduced influence of aerodynamic and mechanical complexities; the turbine operates in a more linear torque control regime, where power production closely follows a predictable cubic relationship with wind speed. As wind speed increases, turbulence and wake interactions introduce greater variability, leading to higher uncertainty in power output and, consequently, a higher standard deviation. However, as the wind speed approaches the rated value, the turbine enters pitch control mode, where power output stabilizes at the rated capacity, reducing variability and lowering the standard deviation again. This approach means that our NBM methodology tolerates higher errors in regions where there is more uncertainty about the potential power and enforces stricter error thresholds in regions where the potential power is more certain. This adaptive error tolerance is crucial for accurately identifying abnormal behavior without generating excessive false positives, particularly in regions where power output uncertainty is naturally higher."

- 1) Line 276-277: Did you have any way of validating or checking the feasibility of the status logs?
   2) These status logs were generated by the SCADA system of which we assume were correct. Still, we did a qualitative check by manually checking some status logs.
   3) n/a
- 1) Line 288: I would start a separate sub-section here (and create a different sub-section for the part above)
  2) We created a new subsection for this part.
  - 3) n/a
- 1) Line 289: How was this period chosen? How representative is it for this turbine?
  2) We added a sentence to clarify that this period was chosen arbitrarily because it includes known loss events. The loss events encountered during this period are representative for the wind farm at hand.

3) "This period was chosen arbitrarily because it includes known loss events, making it suitable for demonstrating the model's ability to detect such anomalies."

• 1) Line 325: I suggest an additional section summarising the results and more broadly discussing the effectiveness and applicability of this method, including other factors (on top of the performance factors studied here) such as computational costs, ease of modelling / training, transferability, etc.

2) We added a separate discussion section where we summarize the main results and discuss the overall effectiveness of our method.

3) "As demonstrated in the Sect. 4, our GNN-based power prediction models consistently outperform traditional methods for both predicting power during normal operations and detecting abnormal events. The Spatio-Temporal GNN's unique ability to incorporate lagged input feature values introduces an additional temporal dimension, enabling it to identify and exploit trends over time. This temporal insight proves particularly valuable in capturing dynamic wind condition changes, such as fluctuations in wind speed and direction, while also smoothing out noise in high-frequency SCADA data. Our findings show that the Spatio-Temporal GNN excels in power prediction, especially under waked conditions. Unlike the power curve binning method and a simple MLP, which struggle to accurately predict energy ratios in these scenarios, the Spatio-Temporal GNN achieves significantly greater accuracy. This consistency between model predictions and the intuitive understanding of turbine interactions under varying wind conditions highlights the model's capability to effectively capture the intricate dynamics of wake interactions within the wind farm. Additionally, we validated our model in scenarios lacking ground truth data by using it to detect anomalies through deviations from expected behavior. The GNN-based methodology demonstrated remarkable proficiency in detecting nearly all power loss events, whereas the power curve binning method failed to identify a substantial portion of these anomalies. The superior performance of the GNN models emphasizes the advantage of

representing the wind farm as a graph. In this representation, each turbine not only considers its local measurements but also incorporates information from neighboring turbines through message passing, capturing complex spatial dependencies across the wind farm. By integrating wind farm topology into power predictions and learning local features, we anticipate that this model is transferable to new and unseen wind farms. However, this claim needs further investigation. Despite its advanced capabilities, developing and training the Spatio-Temporal GNN requires minimal effort. A single model suffices for an entire wind farm, and training time ranges from minutes to a few hours on a GPU, depending on the selected hyperparameters. This efficiency, combined with its predictive power, makes the GNN-based approach a practical and scalable solution for wind farm power prediction and anomaly detection."